# Disorder-induced single-mode transmission

Giancarlo Ruocco[1,2], Behnam Abaie[3], Walter Schirmacher[1,2,4], Arash Mafi[3] & Marco Leonetti[1,5]

Localized states trap waves propagating in a disordered potential and play a crucial role in Anderson localization, which is the absence of diffusion due to disorder. Some localized states are barely coupled with neighbours because of differences in wavelength or small spatial overlap, thus preventing energy leakage to the surroundings. This is the same degree of isolation found in the homogeneous core of a single-mode optical fibre. Here we show that localized states of a disordered optical fibre are single mode: the transmission channels possess a high degree of resilience to perturbation and invariance with respect to the launch conditions. Our experimental approach allows identification and characterization of the single-mode transmission channels in a disordered matrix, demonstrating low losses and densely packed single modes. These disordered and wavelength-sensitive channels may be exploited to de-multiplex different colours at different locations.

[1] Center for Life Nano Science@Sapienza, Istituto Italiano di Tecnologia, Viale Regina Elena, 291 00161 Roma, Italia. [2] Department of Physics, University Sapienza, P.le Aldo Moro 5, I-00185 Roma, Italy. [3] Department of Physics and Astronomy, Center for High Technology Materials, University of New Mexico, Albuquerque, New Mexico 87131, USA. [4] Institut für Physik, Universität Mainz, Staudinger Weg 7, D-55099 Mainz, Germany. [5] CNR NANOTEC-Institute of Nanotechnology c/o Campus Ecotekne, University of Salento, Via Monteroni, 73100 Lecce, Italy. Correspondence and requests for materials should be addressed to M.L. (email: marco.leonetti@iit.it).

The ability to confine waves is of paramount importance in laser physics[1], matter waves[2] and optics communications[3], while it is fundamental in many technological applications such as optical tweezers[4], acoustics[5] or quantum optics[6]. 'Single-mode' confinement is a peculiar form of wave storage, in which a single solution of the wave equation is trapped in a system producing a coherent wavefront, which is not subjected to variations originated by interference with other modes[7–9].

In the case of multimode fibres, light is launched into many modes at once, resulting in a fluctuating intensity profile and in output strongly dependent on the launch conditions and influenced by bending and stretching of the core. This is due to the interference between the modes. Single-mode fibres are today fundamental in communication technologies, allowing extremely high bandwidth data communication immune to intermode dispersion[10,11]. Single-mode operation instead is obtained by properly shaping a symmetric fibre core[7] or with more sophisticated physical effects[8,9] so that a single-mode fibre with 5–10 μm core[12] provides a single oscillating solution. In such a regime, the intensity profile has a fixed shape along propagation and is independent of the launch conditions: the input-coupling conditions only influence the amount of light coupled to the channel. Absence of defects is important for the realization of high-quality single-mode fibres: any unwanted scattering results in leakage or high-order-mode contributions.

On the other hand, disordered systems may provide a form of confinement, which is know as Anderson Localization[13,14]. Anderson Localization, a phenomenon still presenting some unexplored aspects[15], is known to produce a drastic reduction of the diffusion coefficient[16]. In his seminal paper, Anderson has shown[14] that in a strongly disordered system the available modes are localized states[13]: a wave in such a system cannot propagate freely but has to jump from a localized state to another, resulting in a nearly zero diffusion coefficient. Localized states are both spatially separated and energetically distant (they are isolated[17,18]), so that energy has a small probability to be transferred from one state to a neighbour. In other words, if the disorder is strong enough, it may in principle support single-mode states in the form of individual solutions of the wave equation, which do not exchange energy with other degrees of freedom.

Here we report the observation of single-mode states in the regime of Transverse Anderson Localization (TAL) in a disordered optical fibre. TAL is a peculiar form of localization requiring that the refractive index distribution in the transversal plane to the propagation direction must be disordered and the system must be homogeneous along the propagation direction. These two conditions have been experimentally tested for electromagnetic waves in photonic lattices[19] and in optical fibres[20–22], demonstrating that TAL is able to compensate for the natural diffraction of light producing a constant transversal profile size. We exploited disordered binary fibres (DBF)[20], realized by melding micron-sized strands of polymethyl-methacrylate (refractive index $n = 1.49$) and polystyrene ($n = 1.59$), so that the final product is a disordered structure in the $x$, $y$ plane, which is invariant along propagation direction $z$ (refs 23,24). A complete characterization of the disordered fibres is discussed elsewhere[25–28]. The transmission channels sustained by these disordered and longitudinally invariant systems are quite robust[23], and demonstrated with several configurations and materials[19,21]. These systems behave as a coherent fibre bundle: light injected at a certain location exits at the corresponding location at opposite facet[29] and it is possible to exploit simultaneously several transmission channels at different wavelengths in the same fibre[30] (see also Supplementary Note 1).

## Results

**Locating transmission channels.** To locate the transmission channels of a DBF (experimental set-up is shown in Fig. 1, a fibre scheme and image of the tip are shown in Fig. 1c,d), we measured the total transmission as a function of the input position (see Methods). The result is reported in Fig. 2a for a 20-cm-long disordered fibre. $T(\mathbf{r})$ ($\mathbf{R}$ is the output coordinate and $\mathbf{r}$ the input coordinate, see Methods section) is strongly position-dependent ranging from $10^{-3}$ to nearly unity (the distribution reported in Fig. 2f as blue triangles); moreover, the high transmission areas appear to be organized in a set of sparse and highly transmitting channels embedded in a scarcely transmitting sea. A single scan is visualized in Supplementary Movie 1 for a DBF, while in Supplementary Movie 2 we report the same measure for a standard homogeneous multimode fibre. Polarization properties of the modes are investigated in the Supplementary Note 2. The first clue on the single-mode nature of these channels comes from the pattern of the transmitted intensity $I(\mathbf{R}, \mathbf{r})$, which is not affected by a small displacement of the input: $I(\mathbf{R}, \mathbf{r}) = I(\mathbf{R}, \mathbf{r} + \Delta\mathbf{r})$ if $\Delta\mathbf{r}$ is small and $\mathbf{r}$ is the location of the localized mode. The two profiles obtained for the input in

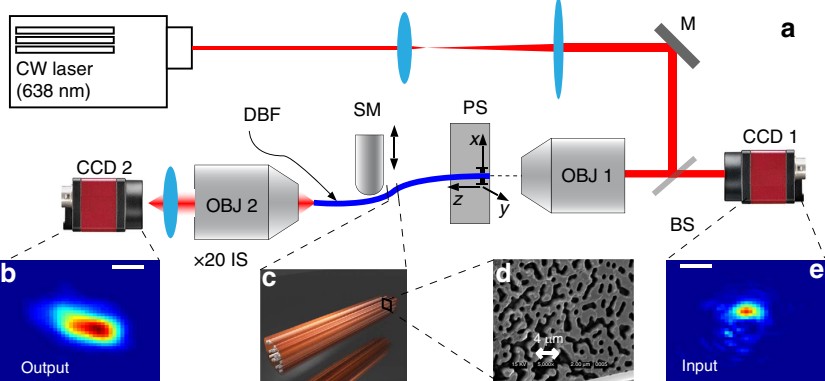

**Figure 1 | Experimental setup.** (**a**) A light beam from a CW 674 nm laser is expanded and focused on the disordered fibre (DBF, **c** reports a 3D sketch of the fibre; **d** reports the black and white scanning electron microscope image showing the fibre facet detail) by a long working distance Objective (OBJ 1). Back reflected light from the fibre input is imaged on a CCD1 (charge-coupled device camera; **e** reports a typical input light spot; scale bar, 1 μm). The fibre input is mounted on a X,Y piezo actuator, which allows a fine control of the input position ($20 \times 20$ μm$^2$ 5 nm spatial resolution). The output (typical output in **b**), scale bar, 1 μm) of the fibre is collected by a second objective (OBJ 2), which is part of a $\times 20$ imaging system aligned to CCD 2 (**e** reports a typical output light spot).

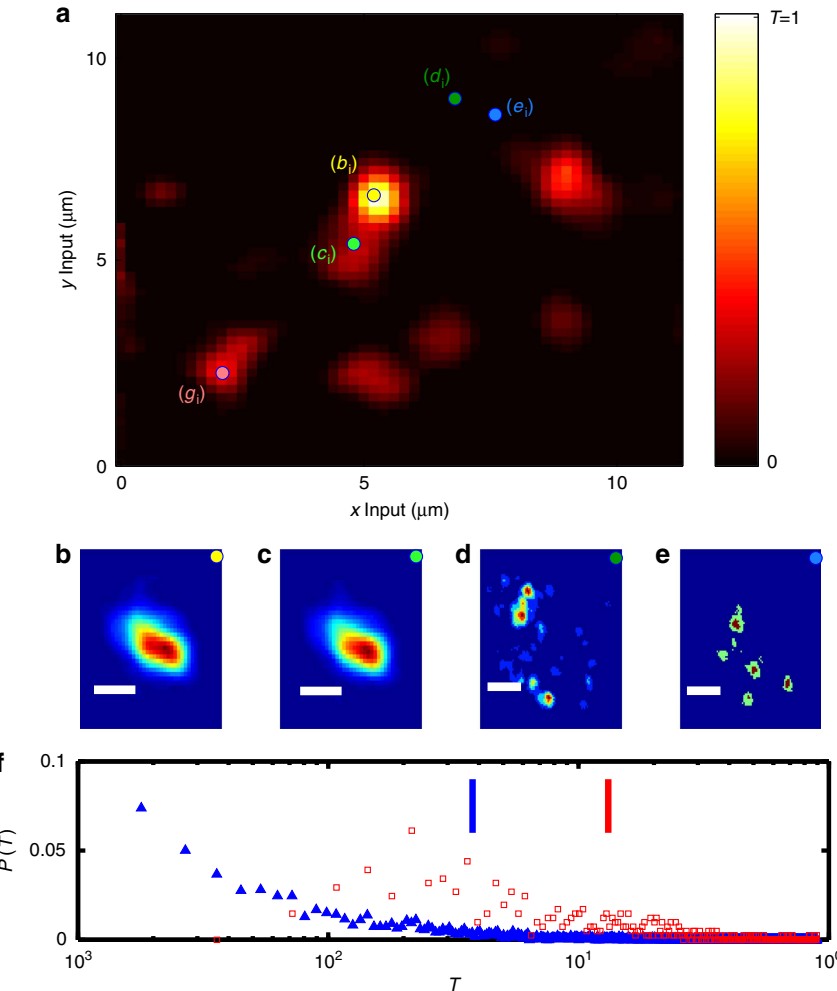

**Figure 2 | Transmittance properties.** (**a**) Total transmittance as a function of the input location. Relevant input locations are indicated with coloured full circles. (**b**–**e**) Output patterns retrieved for different input locations indicated by the relative coloured markers and labelled as '$b_i$,' '$c_i$,' '$d_i$,' '$e_i$'. (**f**) Probability density function for the transmittance obtained from 3,136 different input locations. Blue (red) vertical line indicates the average transmittance for all the measured locations (only for high $Q_{AB}$ ($Q_{AB} > 95\%$) input locations). Scale bar, 2 μm (**b,c**) and 4 μm (**d,e**).

correspondence of the maximum transmittance (yellow dot in Fig. 2a labelled as $b_i$) and for an input slightly shifted (green dot in Fig. 2a labelled as $c_i$) are nearly indistinguishable. On the other hand, the intensity profile obtained for a barely transmitting input area provides a strongly varying output pattern if the input location is slightly displaced (see Fig. 2d,e where speckles obtained for inputs at $d_i$ and $e_i$ are shown).

In other words, in the neighbourhood of a highly transmitting channel, the shape of the output intensity remains almost unperturbed, exactly like in a single-mode fibre where the shape of the transmitted wavefront is independent on the launch conditions. We now proceed to directly quantify the degree of invariance of the light profile transmitted by a transmission channel. To establish the degree of similitude of two intensity distributions obtained for two different inputs $\mathbf{r}_1$ and $\mathbf{r}_2$, we will exploit the observable $Q(\mathbf{r}_1, \mathbf{r}_2)$ (see Methods) to build $Q$-maps.

**$Q$-maps.** We are interested in finding regions in which the shape of the output pattern does not depend on the launch conditions that is where $Q(\mathbf{r}_1, \mathbf{r}_2) \cong 1$: the invariance of the output profile is a signature of the single-mode transmission. We expect that the single-mode transmission channels are found in correspondence of a transmittance maxima (for example, ($b_i$) with the profile

shown in Fig. 2b); therefore, we computed $Q(\mathbf{r}_{b_i}, \mathbf{r}_2)$ for all inputs $\mathbf{r}_2 = [x_2, y_2]$. The result is shown in Fig. 3a. The white area represents the inputs $\mathbf{r}_2$ for which $Q(\mathbf{r}_{b_i}, \mathbf{r}_2) \cong 1$, that is where the output patterns are very similar. This result indicates that if light is injected close to $\mathbf{r}_{b_i}$ the output profile and shape do not vary, that is, that the mode at $\mathbf{r}_{b_i}$ is unaffected by the launch conditions.

The same procedure has been repeated for the mode at $r_{g_i}$ for which we retrieve the $Q(\mathbf{r}_{g_i}, \mathbf{r}_2)$ profile reported in Fig. 3b, showing that $g_i$ is immune to the launch conditions close to $r_{g_i}$. As a comparison in panel Fig. 3c we report $Q(\mathbf{r}_{d_i}, \mathbf{r}_2)$, with $r_{d_i}$ corresponding to a poorly transmitting input that is sensible to the launch conditions. To quantify the degree of isolation of a mode, we introduce the 'dwelling area' (DA) of a mode (the area of the input that allows to couple all injected light to the same mode), as the area for which $Q > 0.9$. Mode $b_i$ results in a $DA = 8 \mu m^2$, mode $g_i$ has $DA = 1 \mu m^2$, while for $d_i$ $DA = 0.04$ $\mu m^2$. By scanning 20 different regions of $100 \mu m^2$ we found an average of 6.5 single modes with $DA > 0.5 \mu m^2$. This means that a DBF, which is $250 \times 250 \mu m^2$, is able to host $\sim 4,000$ single modes.

**Fibre bending.** Another requirement for single-mode transmission is the resilience to fibre bending. If transmission is carried

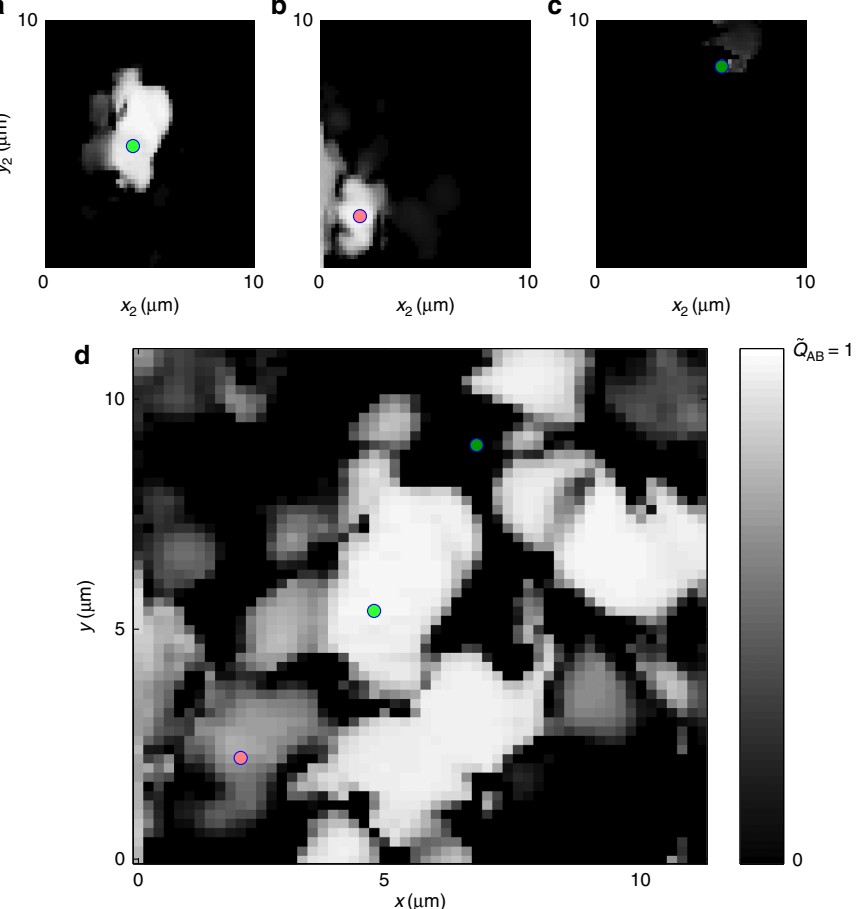

**Figure 3 | Correlation maps.** (**a–c**) $C_1$ $C_2$ and $C_3$ of modes in $b_i$, $g_i$, $d_i$ (as labelled in Fig. 2a). In **d** the total correlation $C$ obtained by displacing the fibre by 5 mm near its centre is reported.

out by a single-mode channel, interference is absent so that the transmitted intensity profile is not affected by the fibre folding. To check this feature, we exploited an actuator capable of displacing a 20 cm fibre by 1 cm (Sample Mover (SM) in Fig. 1) from a starting configuration (configuration A) to a second one (configuration B). Then, we compared the output intensity profile in the two configurations and computed $Q_{AB}(\mathbf{r}) = \int I_A(\mathbf{r}, \mathbf{R}) I_B (\mathbf{r}, \mathbf{R}) d\mathbf{R}$ for all inputs $\mathbf{r} = [x, y]$. $\tilde{Q}_{AB}(\mathbf{r})$ reports the degree of similitude of intensity patterns with the same input but in two different fibre configurations. If $\tilde{Q}_{AB}(\mathbf{r})$ is close to 1, the fibre displacement is not affecting the light coupled to the fibre at that position, so that $\tilde{Q}_{AB}$ is a measure of the modes' resilience to fibre bending. The result is reported in Fig. 3d, where the clear areas ($\tilde{Q}$ close to 1) are due to the presence of bending-resistant transmitting channels, while darker areas correspond to the absence of a single mode at the input location wavelength. In that case, light undergoes diffusion before being coupled to a guided mode or is expelled by lateral leakage yielding an attenuated output, which is strongly dependent on the input (see Supplementary Note 3).

By comparing $\tilde{Q}_{AB}(\mathbf{r})$ with $T(\mathbf{r})$ in Fig. 2a, we notice that the higher the fibre transmittance, the higher the resilience to fibre bending: thus, the resilience and the transmittance are correlated. This is proved in Fig. 2f where we report the distribution of transmittances retrieved from the scan reported in Fig. 2a with full triangles, while the transmittances for the locations with $\tilde{Q}_{AB} > 0.95\%$ (high resilience to fibre bending) are reported by open squares. The average transmittance $\langle T \rangle$ is 0.04 (visualized by a blue vertical line in the graph), while the average

transmittance for the resilient areas $\langle \tilde{T} \rangle$ is 0.13 (visualized by a red vertical line in the graph).

**Average properties.** In sum, it is possible to inject light in a disordered fibre in order to feed modes with intensity profiles that are independent on the launch conditions and resilient to fibre bending; moreover, these modes are located at the transmission maxima and provide a higher transmittance with respect to the average. Now we want to prove that all these characteristics are independent on the particular realization of disorder, and may be found for any disorder realization. In Fig. 4a we report $\langle T \rangle$ (blue squares) and $\langle \tilde{T} \rangle$ (red circles) obtained by averaging five different disorder realizations obtained from these different fibres (lengths $L = 4$, 10 and 20 cm) for a total of 15 disorder realizations analysed. The transmittance decreases exponentially as $C_0 \exp^{-T(L)/\ell}$, where $C_0$ is a parameter for the coupling efficiency and $\ell$ is the length over which the transmittance decays of a factor $e$ (leakage length). By fitting the data, it is possible to find the average leakage length $\ell_A = 7$ cm and the leakage length for the single-mode locations, which is much higher $\ell_C = 230$ cm. Results presented in Fig. 4a) are retrieved from 15 different measurements confirming the generality of our result.

Now we report on the response of the system at different wavelengths. At difference with standard single-mode optical fibres, which are usually broadband, disorder hosts modes with a narrow linewidth[31] and this property has been recently exploited for applications[32,33]. In addition, disordered single modes are sharp, possessing a subnanometric linewidth[34], so that the

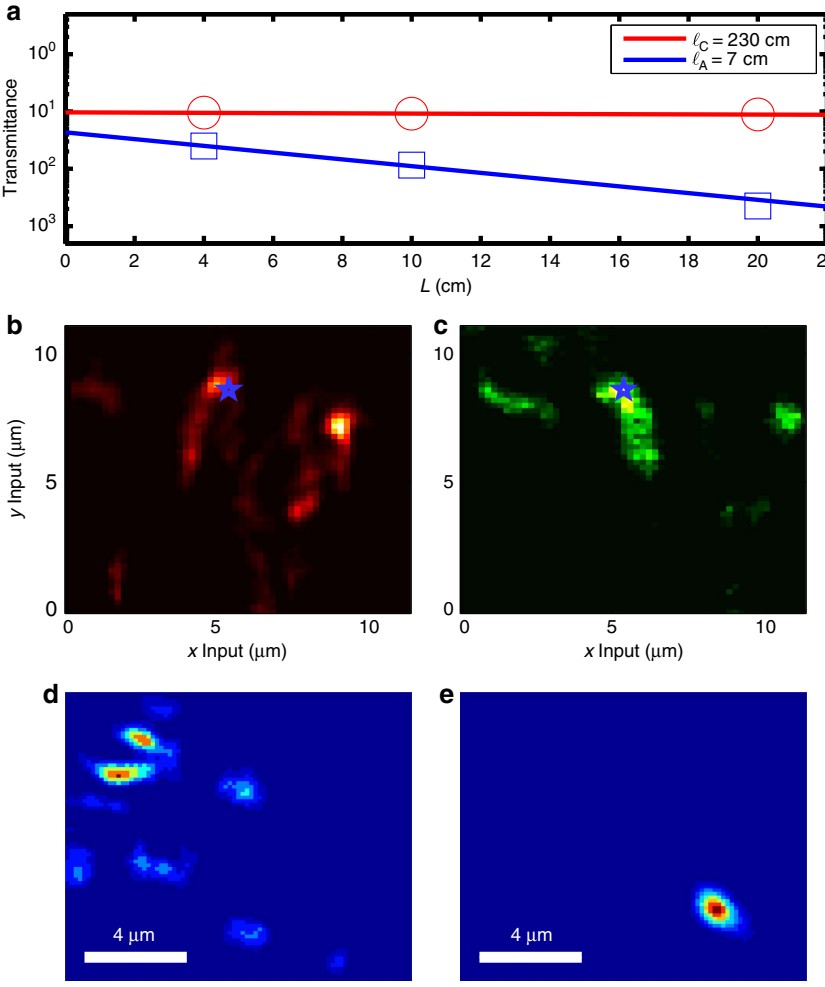

**Figure 4 | Length and wavelength analysis.** (**a**) Average transmittance for highly correlated inputs (red circles) as a function of fibre length. Losses for single-mode coupling are of $2\,dB\,m^{-1}$ (red circles). Average losses are of $62\,dB\,m^{-1}$ (blue squares). (**b**,**c**) Report the transmittance maps obtained, respectively, with a 638 nm input and with a 532 nm input. (**d**,**e**) Report the intensity profile at the fibre output tip for the two wavelengths (638 and 532 nm, respectively) for the input at $r_{star}$, location indicated with a star in **b**,**c**. Scale bar, 4 μm (**d**,**e**).

transmittance of the system strongly depends on the wavelength. In Fig. 4b,c the transmittance maps are obtained at 674 and 532 nm. The locations of the transmittance maxima are different, and the output intensity profiles change. We fixed the input at the location (indicated the transmittance maps (Fig. 4b,c), with a star marker) and reported the two intensity outputs in Fig. 4d,e. The two profiles are nonoverlapping so that if light is injected at $r_{star}$ the two colours are de-multiplexed.

## Discussion

In summary, modes in a strongly disordered system show a negligible crosstalk accompanied by a inherent resilience to fibre bending and an invariance with respect to the launch conditions. These single modes show a higher transmittance with respect to the locations in which single modes are absent. We also exploited the narrow linewidth of these modes to obtain separation of different colours at different localized output inside the DBF.

## Methods

**Experimental details.** To locate highly transmitting channels, we exploited a high numerical aperture (NA) objective (EDMUND #59–880 long working distance objective, NA 0.8) to focus light at the fibre's input (see Fig. 1e) and a precision piezo stage, which allows to slightly move (piezo scanning range $20 \times 20\,\mu m^2$) the fibre input in order to scan different input locations. Therefore, the intensity $I$ at the fibre output at coordinates $[X, Y] = \mathbf{R}$ depends on the input illumination position $[x, y] = \mathbf{r}$: $I(\mathbf{R}, \mathbf{r})$. The total transmittance as a function of the input

position is $T(\mathbf{r}) = \sum_{\mathbf{R}} I(\mathbf{R}, \mathbf{r})/I_{in}$, where $I_{in}$ is the total injected intensity as measured by CCD1.

**Q-maps.** $Q(\mathbf{r}_1, \mathbf{r}_2) = \int \tilde{I}(\mathbf{r}_1, \mathbf{R}_o)\tilde{I}(\mathbf{r}_2, \mathbf{R})d\mathbf{R}$ where $\tilde{I}(\mathbf{r}, \mathbf{R})$ represents the intensity $I(\mathbf{r}, \mathbf{R})$ normalized in such a way that $Q(\mathbf{r}, \mathbf{r}) = 1$. $Q$ represents the degree of similarity of the intensity profiles at the output for the inputs at $\mathbf{r}_1$ and $\mathbf{r}_2$: it is close to 1 if the output patterns are identical, it is close to zero if light intensity is located in two nonoverlapping regions, while it is close to 0.5 for extended random speckle patterns.

**Data availability.** The data that support the findings of this study are available from the corresponding author upon request.

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

## Acknowledgements

M.L. acknowledges 'Fondazione CON IL SUD', Grant 'Brains2south', Project 'Localitis'.

## Author contributions

M.L. performed experiments; M.L. and G.R. designed the experiment; A.M. and B.A. fabricated the sample. All the authors contributed in writing the manuscript and interpreting the data.

## Additional information

**Competing financial interests:** The authors declare no competing financial interests.

**Publisher's note**: 

