## [Peer Review File · Nature Communications]

Reviewers' comments:

Reviewer #1 (Remarks to the Author):

The manuscript studies experimentally single-mode transmission in a disordered optical fiber due to transverse Anderson localization of light. The main potential value of the disordered fiber is based on the demonstration that it supports several regions of single-mode transmission. Authors point out potential applications in optical communications, quantum optics, tweezers, etc. However it is doubtful if this fiber can be used in practice for such applications:

1) Simultaneous transmission of multiple signals through the single-mode sites has not been demonstrated. It seems that this could maybe be achieved only by resorting to complex and expensive set-ups involving spatial light modulators to perform beam shaping for both in- and out-coupling, but that seems extremely challenging for applications (especially since each fiber is different, and would need to be characterized separately).

2) The observed coupling efficiency to the modes was quite low, about 10% or less, while the leakage length is just 230cm. With respect to previous comment, it may be expected that efficiency may drop even further for multiple mode transmission. However, much better transmission can be achieved with multi-core and photonic-crystal fibers over such distances.

3) In terms of fundamental physics the authors acknowledge that "Localized states are both spatially separated and energetically distant (they are isolated 17, 18)", indeed the existence of single-mode transmission regions is fully expected based on previous theoretical and experimental demonstrations.

4) The polarization properties of the light transmission are important in fiber links, but these are not analysed in the manuscript. Does each single-mode site transmit both polarizations? Is there polarization rotation? What is the response to bending and/or twisting of the fiber?

5) The transmission is demonstrated to be different at the wavelengths which are quite far apart, 640nm and 532nm. It is not discussed why these particular wavelength combinations may be useful, and their simultaneous in/out coupling was not demonstrated. A detailed analysis of the spectrum transmission over a particular wavelength range would provide more useful information, in particular regarding the bandwidth of single-mode transmission channels, mode dispersion, etc.

Overall, the manuscript presents the observation of physical effects which are fully expected based on multiple preceding theoretical and experimental studies, while the degree of technical advances does not seem appealing for applications.

Reviewer #2 (Remarks to the Author):

Very nice paper in which, experimentally, a disordered structure is exploited to realise single mode fiber transmission. The authors manage to make an optical fibre that is (truly very much) disordered in the two lateral directions, while being perfectly translationally invariant along the third dimension. This as such is already quite an experimental feat.

The authors then show results of optical transmission measurement where they manage to identify modes that are separated in frequency from other modes, and that are robust against small changes in incoupling conditions and fiber bending. They then convincingly show to have found single mode transmission channels, due to transverse localisation of light.

I find the results convincing and of broad interest. This is a very interesting application of disordered photonics that might make it into the Thorlabs catalog on day, alongside photonic crystal fibres. Can be published in my view in Nature Communications.

Reviewer #3 (Remarks to the Author):

The manuscript by Leonetti et al. studies transverse Anderson localization in random fibers. The central point of their study is the observation of single mode transmission as evidenced by a lack of sensitivity to input parameters.

The results are interesting and of value to the community of multiple scattering. I see two main issues with the manuscript:

1. The manuscript lacks details on the nanofiber fabrication. This is not only a question of experimental details but more importantly about the quantitative measures of disorder in the fibers. Without this information, it is hardly possible to reproduce the results and even less possible to understand the implications of the author's findings, e.g., to what extent is this always the case? Another aspect is to what extent the fibers are in fact translationally invariant along the fiber axis, which is a crucial underlying assumption that appears unjustified.

2. The statistical analysis appears sound but the underlying electromagnetic problem is not touched upon. This relates to point 1 above. By numerically analyzing the transmission for different disorder realizations it would be straightforward to identify under what conditions the transmission would be single-moded or multi-moded. This would not only clarify the underlying physics but also the generality of the observed effects. In the present version, it is not clear if the observations are particular to one batch of random fibers or if they are in fact of a more general nature.

I would likely recommend publication of a revised manuscript with the above mentioned issues properly addressed.

Reviewers' comments:

Reviewer #1 (Remarks to the Author):

The manuscript studies experimentally single-mode transmission in a disordered optical fiber due to transverse Anderson localization of light. The main potential value of the disordered fiber is based on the demonstration that it supports several regions of single-mode transmission. Authors point out potential applications in optical communications, quantum optics, tweezers, etc. However it is doubtful if this fiber can be used in practice for such applications:

We thank the referee for inspecting and evaluating our manuscript. We strongly believe that our system shows several advantages with respect to standard single mode optical fiber bundles. In the present version we responded to all raised issues and clarified the advantages of a disorder-based approach.

1) Simultaneous transmission of multiple signals through the single-mode sites has not been demonstrated. It seems that this could maybe be achieved only by resorting to complex and expensive set-ups involving spatial light modulators to perform beam shaping for both in- and out-coupling,

In the present version, we added in the supplementary materials the details of multiple wavelength experiment. In the figure below we show experimental setup

We added all the information in the "Multi-mode operation" subsection in the supplementary materials, together with a detailed description of the experiment. There we show that there is no need for complex beam shaping of spatial light modulator but a simple beam splitter or dichroic mirror allows for simultaneous coupling of multiple frequencies to the disordered transmission channels. Similar results have been demonstrated previously in the reference [Multiple-beam propagation in an Anderson localized optical fiber, Opt. express 21, 305,] which has been added to the main paper's references (ref 30).

Moreover we stress that, in general, localized states have a complex shape of the wave front and a perfect matching between the input light and the mode shape would require a complex shaping of the wavefront.

Our approach instead exploits a focused laser beam which, in general, does not match the mode shape, providing a non-optimal coupling. This reduction of efficiency however does not affect the degree of isolation as soon as the input beam is smaller than the mode's dwelling area.

but that seems extremely challenging for applications (especially since each fiber is different, and would need to be characterized separately).

1) The fabrication process (drawing tower) enables to produce very long fibers, (3 km length) with a single disorder realization. These fibers may be individually characterized and then separated in many meter-sized pieces. Indeed for industrial/commercial issues, one has to characterize just one of the single long fiber piece in order to get the full characterization for the whole set.

Therefore we do not see technical/fundamental impediments, in producing pre-characterized optical fibers which may be sold together with a database indicating the optical properties and mode maps.

2) Even without characterization, the number of single modes hosted by the fiber is very large (see below), so that a user may easily find one (or more see Multi-mode operation) single mode transmission channels at a random location. If the user does not need to exploit a particular mode, transmission may be performed easily just by picking a random one.

3) The fiber may be also used without targeting a single mode but just exploiting all the transmitting channels in an area (see reference [25, 26,28] of the main paper) , or even to transport images (see reference [29] of the main paper).

2) The observed coupling efficiency to the modes was quite low, about 10% or less, while the leakage length is just 230cm. With respect to previous comment, it may be expected that efficiency may drop even further for multiple mode transmission. However, much better transmission can be achieved with multi-core and photonic-crystal fibers over such distances.

The referee is right when he states that at the present day, our fiber has a transmission efficiency much lower than the silica/plastic multi mode fibers, however:

I) we know that the majority of the losses in our system is due to impurities and may be overcome at fabrication stage, thus achieving the same transmittance of standard plastic optical fibers.

II) The number of single modes hosted by our system is larger than that supported by standard multi core bundles. Thus providing a clear advantage on this specific aspect, versus standard fiber technology.

I) The losses of a single mode system are due to two main features, core Absorption/Scattering and overlap of the mode tails with the fiber's exterior. Due to the fact that the modes have a transverse size two orders of magnitude lower than the fiber size, the losses are primarily due to scattering/absorption from the plastic materials. We believe that an optimization of the fabrication process (similar to that which has been performed for silica fiber or photonic crystal fibers in the last decades), may drastically improve the fiber transmittance by reducing the amount of scatterers/impurities. Our fabrication process has not been optimized (this is a proof of principle demonstration), while there are a large set of degrees of freedom which may be studied to improve transmission (drawing temperature, ambient presence of dust and dirt, fiber handling automation, materials etc..), which may bring the transmittance to the same level of the standard multi mode plastic optical fibers.

II) Current single mode fiber bundles rely on the careful fabrication and placement of individual single mode fibers, at a distance which does not produce inter mode cross-talk. Our approach does not need such effort at the fabrication stage: the disordered arrangement of plastic strands supports well separated and non-coupled single modes without the need of individually designing every single fiber core.

Without effort on the design, the binary fiber hosts about 4000 single modes (at 670 nm which) which is comparable, with what found in literature for "designed" multi core fibers (for example <http://myriadfiber.com/fujikura-image-fiber-quartz-glass/>, which are not NOT single mode, because the cores are coupled between each other).

In other words our system provide huge number of single mode transmission channels with a much lower effort at the fabrication stage. We believe that this unique possibility may be relevant for applications.

We stressed this point in the text adding the following sentence:

"To quantify the degree of isolation of a mode we introduce the "dwelling area" (DA) of a mode (the area of the input that couples just to a given mode), as the area for which $Q > 0.9$. Mode b_i results in a $DA = 8 \mu\text{m}^2$, mode g_i has $DA = 1 \mu\text{m}^2$, while for d_i $DA = 0.04 \mu\text{m}^2$. By scanning twenty different $100 \mu\text{m}^2$ regions we found an average of 6.5 single modes with $DA > 0.5 \mu\text{m}^2$. This means that a DBF, which is $250 \times 250 \mu\text{m}^2$ is able to host about 4000 single modes. "

3) In terms of fundamental physics the authors acknowledge that "Localized states are both spatially separated and energetically distant (they are isolated 17, 18)", indeed the existence of single-mode transmission regions is fully expected based on previous theoretical and experimental demonstrations.

We agree with the referee on the fact that the physics of disorders figures out the existence of "isolated" single modes: the impossibility for energy to travel from a mode to another is at the basis of the Anderson localization phenomenon.

On the other hand we totally disagree on the emphasis that the referee gives to this statement because the idea to exploit these "isolated modes" to generate single modes similar to that hosed inside optical fibers is totally new for the localization/disorder community because it was never featured or envisaged in previous papers.

Anderson localization is known from more than 50 years and transverse localization has been demonstrated experimentally since about 10 years and yet we was not able to find a reference to the possibility to exploit disorder to produce "single mode" transmission channels.

In particular up to now it has never been demonstrated that Anderson localization produces modes which are "resilient" to changes of the input conditions, such as the ones of single mode fibers.

This aspect has never been proposed and here we demonstrate it experimentally.

We stress that the third referee states expressly that the result is interesting for the multiple scattering community. We agree with him because disorder induced single modes bridge two communities which are far away (disorder and optical communications) for the first time, and we believe that this is already a huge result.

Moreover we also provide a novel experimental technique which enables to recognize single modes and identify their position: it is possible for the first time to unravel the nature of modes in a complex disordered landscape.

We believe that our results change the perspective for disorder: from a source of disturbance for transmission to an platform able to sustain a refined and convenient form of communication.

4) The polarization properties of the light transmission are important in fiber links, but these are not analyzed in the manuscript. Does each single-mode site transmit both polarizations? Is there polarization rotation? What is the response to bending and/or twisting of the fiber?

We included in the present version of the paper a characterization of the linear polarization properties of the fiber. We measured the fiber response as a function of input polarization in both the polarization maintaining channel (Vertical-Vertical and Horizontal-Horizontal) and in the Vertical- Non polarized channel.

Experimental setup and results for the transmittance maps for the various polarization configurations are shown in the figure below

which is also reported in the "polarization properties" section in the supplementary materials, which includes measurements details and results for fiber to twisting and bending.

5) The transmission is demonstrated to be different at the wavelengths which are quite far apart, 640nm and 532nm. It is not discussed why these particular wavelength combinations may be useful, and their simultaneous in/out coupling was not demonstrated. A detailed analysis of the spectrum transmission over a particular wavelength range would provide more useful information, in particular regarding the bandwidth of single-mode transmission channels, mode dispersion, etc.

The referee is concerned by the specific set of wavelengths used. At difference with silica fibers, plastic optical fibers are designed and engineered for visible applications. The aim of experiments with two different wavelengths is to demonstrate the strong sensitivity to wavelength typical of disordered systems. A disordered structure produces in fact sharp resonances, and this sharpness is known to be at the basis of the Anderson localization (sharp resonance produce small frequency overlap so that modes become isolated). The wavelength response of a DBF is completely different to that of a standard single mode fiber which is usually broadband. Our two-wavelengths measurement is aimed to emphasize this difference: the two wavelengths give rise to two completely different transmittance maps, while the set of transmitting modes is completely different.

To clarify this point we added the following sentence to the manuscript:

"At difference with standard single mode optical fibers, which are usually broadband, disorder give rise to modes with a narrow line width [26] and this property has been recently exploited for applications [27, 29]."

A more refined analysis of the spectral properties of disordered fibers may be found in previous papers (reference [31]).

[Redacted]

Overall, the manuscript presents the observation of physical effects which are fully expected based on multiple preceding theoretical and experimental studies, while the degree of technical advances does not seem appealing for applications.

Having responded to all the points raised, we hope that now the referee can consider our paper feasible for publication.

Reviewer #2 (Remarks to the Author):

Very nice paper in which, experimentally, a disordered structure is exploited to realize single mode fiber transmission. The authors manage to make an optical fiber that is (truly very much) disordered in the two lateral directions, while being perfectly translationally invariant along the third dimension. This as such is already quite an experimental feat.

The authors then show results of optical transmission measurement where they manage to identify modes that are separated in frequency from other modes, and that are robust against small changes in coupling conditions and fiber bending. They then convincingly show to have found single mode transmission channels, due to transverse localization of light.

I find the results convincing and of broad interest. This is a true application of disordered photonics that might make it into the Thorlabs catalog on day, alongside photonic crystal fibers. Can be published in my view in Nature Communications.

We thank the referee for the fully positive referral, and for acknowledging the degree of novelty of our results.

Reviewer #3 (Remarks to the Author):

The manuscript by Leonetti et al. studies transverse Anderson localization in random fibers. The central point of their study is the observation of single mode transmission as evidenced by a lack of sensitivity to input parameters.

The results are interesting and of value to the community of multiple scattering.

We thank the referee for the positive referral and for recognizing our paper as interesting. Below we address all the points raised.

I see two main issues with the manuscript:

1. The manuscript lacks details on the nanofiber fabrication. This is not only a question of experimental details but more importantly about the quantitative measures of disorder in the fibers. Without this information, it is hardly possible to reproduce the results and even less possible to understand the implications of the author's findings, e.g., to what extent is this always the case?

The referee asks about the amount of disorder and repeatability of the results we are reporting here. About the disorder: it has been completely characterized in a set of previous papers from Arash Mafi and coworkers (we added references 23, 24, 26, 27, 29 to the present version of the main paper).

In particular the paper "Detailed investigation of the impact of the fiber design parameters on the transverse Anderson localization of light in disordered optical fibers" OPTICS EXPRESS, 20, 18692 (2012), reference 23, directly addresses the effects of different disorder parameters on the degree of the localization.

Moreover we stress throughout the manuscript we analyzed a total of 15 different optical fibers realized from two different batches (completely different disorder realization) of production, with slightly differing fabrication parameters. We do not noticed difference on the average degree of localization in those samples thus confirming the robustness of the localization to variations of the structural parameters.

Another aspect is to what extend the fibers are in fact translationally invariant along the fiber axis, which is a crucial underlying assumption that appears unjustified.

The referee correctly points that one of the requirements for transverse Anderson localization is the longitudinal invariance. In our fabrication process, the large draw ratio guarantees that refractive index profile remains unchanged along the fiber. The proof refractive index invariance for the fibers under study is reported in figure 3 of reference 23.

[Redacted]

Right and left columns of the figure show the same x- y location of the fiber at the input and output end tips of the same fiber. The disorder distribution shown is almost identical.

Moreover as shown in Nature Communications 5, 3362 (2014), the disorder optical fibers work as a coherent fiber bundle capable to transport images. Image transport is extremely sensible to longitudinal disorder and would be severely impaired by misplacement or cross over of the individual disordered strands. The fact that instead image transport works perfectly in the disordered fibers further confirms the transverse Anderson localization regime.

2. The statistical analysis appears sound but the underlying electromagnetic problem is not touched upon. This relates to point 1 a above. By numerically analyzing the transmission for different disorder realizations it would be straightforward to identify under what conditions the transmissino would be single-moded or multi-moded. This would not only clarify the underlying physics but also the generality of the observed effects. In the present version, it is not clear if the observations are particular to one batch of random fibers or if they are in fact of a more general nature.

The referee is concerned by the origin of the residual multi mode propagation we measure in our system. The question is important and justified because in the Anderson regime one always aspects to deal with single modes: one mode for each position/frequency. Here we provide the experimental evidence that the multi-mode propagation is due to residual diffusion at the fiber's beginning.

As shown in Fig. 2(a) in our scan of the input fiber facet, there are large areas at which we are not able to couple efficiently light to the fiber. In those cases light does not vanish: it is either reflected, coupled to lateral leakage or coupled to a guided mode after a relatively long diffusion at the fiber beginning. In the third case, light still arrives (attenuated) at the fiber output, but the initial diffusion makes the output strongly dependant on the input conditions.

The fact that before the onset of localization the field undergoes a standard diffusion. This is well known and acknowledged also in the seminal paper of De-Raedt and coworkers: By looking at Fig. 2 of the seminal paper from De Raedt and coworkers(ref 22 in the main paper), it is possible to notice an expansion in the first quarter of the fiber length. After the initial diffusion, there is the onset of localization and the wave field lateral size is stationary.

The effects of this initial diffusion are unveiled by measurements of the Q-maps of a single mode with varying input numerical aperture (see newly added "**Effects of the input numerical aperture**" section in the supplementary materials). With a small input numerical aperture, just a single mode is activated. With a larger numerical aperture light ,which is not coupled to the single mode, produces a background which is masking the single-mode behavior. This effect is due to trajectories which are coupled to guided modes just after a very long diffusion.

The white sharp area in the first panel on the left (N.A. =0.24) indicates a well isolated single mode. On the contrary the blurred boundaries and the lower Q values in the last map on the right (N.A. =0.75) indicate strong dependence on the input conditions. The degree of isolation is quantified by Dwelling Areas (D.A.) of the same mode (panels in Fig 4 a), which decreases from $8 \mu\text{m}^2$ to $0.75 \mu\text{m}^2$ confirming that the high numerical aperture introduces non guided light rays which decrease the measured D.A. .

We stress that even with very large N.A. the single modes are still there, (input condition does not perturb the mode structure of the fiber), however a large part of the light injected in the fiber is not coupled to it and undergoes diffusion before being coupled to a guided mode. These "localized-after-diffusion" channels, are strongly input-dependent and reduce the effectiveness of our technique to measure the mode's dwelling area.

All these data have been included in the section "**Effects of the input numerical aperture**" section of the supplementary material, which details all the experiments.

By numerically analyzing the transmission for different disorder realizations it would be straightforward to identify under what conditions the transmissino would be single-moded or multi-moded.

The referee suggests to include numerical simulation to clarify the origin of multi mode signal. Having provided a consistent explanation, we do not believe in the added value of a simulation while it would make the paper less readable (larger methods, protocols and explanations needed). Moreover a large set of numerical simulation addressing the electromagnetic problem exactly on our system, have been already previously published by Mafi and coworkers: these references **[23; 25; 27,28]** have been now included in the present version of the main paper.

Regarding the generality of our results, we stress that the same results have been retrieved in fibers with different disorder realization, realized with in different fabrication facilities and fabrications run, with different materials (for example glass/air holes), and with slightly different fabrication protocols.

We added the following sentence and relative references to the paper:

"Transverse Anderson localization in these longitudinally invariant system, is quite robust²³, and demonstrated in several systems^{19, 21}."

I would likely recommend publication of a revised manuscript with the above mentioned issues properly addressed.

We thank the referees for the broad and extensive analysis of our manuscript. We hope it now can be considered feasible for acceptance.

Sincerely

Marco Leonetti.

REVIEWERS' COMMENTS:

Reviewer #1 (Remarks to the Author):

The revision properly addresses all the reviewer questions, and the manuscript can now be recommended for publication.

Reviewer #2 (Remarks to the Author):

Indeed the results of the paper are not competitive on the level of final applications, as remarked by referee 1. However, the underlying physics is not trivial in my view, and the paper makes a significant contribution in this respect. While it is true that pure 2D localisation, and transverse localisation are understood for ideal systems, this is not the case for the complex realisation in this paper, which involves real disordered fibres. In that respect the paper provides a significant contribution to the understanding of the physics involved. Also I would not exclude that competitive applications (that is with high performance markers) will come out at a certain stage. I think this is not the responsibility of the authors at this stage.

On the other hand I find the points raised by referee 3 very valid. I would also like to see these points properly addressed, since they would enhance the value of the paper in a significant way. This holds both for the conceptual questions, as well as the technological ones regarding the fiber structure.

Reviewer #3 (Remarks to the Author):

The revised manuscript addresses the main concerns raised in my previous report as well as the issues raised by the other reviewers. A notable exception is the comments by Reviewer 1 who raises concerns about the application potential. In my opinion, this whole discussion is a bit besides the point:

Reviewer 1 points out that the application potential for quantum optics, tweezers, etc. is likely rather limited and I think Reviewer 1 is obviously right. Multiple-scattering effects have widespread applications whenever ensemble-averaged values are of technological value, cf. white paint and metal wires for electrical conduction, which are arguably huge commercial successes. When it comes to applications where the single realizations are important, such as the single modes in the optical fibers considered here, however, it is very difficult to see that disordered structures would have any virtues whatsoever, and to my knowledge there are no examples in history. For example, a disordered optical fiber with many modes can only be exploited if the transceivers in both ends would be equipped with complex and expensive spatial light modulators. Properly designing the fiber would be a much more realistic approach. This point is also acknowledged by the authors in their response to Reviewer 1. As such, I find the author's response somewhat in denial of facts. It is a mystery to me why the authors try and argue against the obvious but such unrealistic views on applications are unfortunately quite common in science in general, and in the field of multiple scattering in particular.

Having said that, the puzzling thing is that the author's claims in the manuscript are much more balanced and well-argued, and in fact they do not claim that their fibers would have a great application potential, they merely point out that confining waves is of great technological importance, which is unquestionably true. However, the authors do clearly allude to applications, in particular in the introduction. I would consider that putting their work in context.

In short, I agree with Reviewer 1's statements about the lack of application potential, I find the author's response quite like overselling, but I also find the whole discussion a bit besides the point.

The relevant question is if the manuscript is suitable for publication and I find it timely and interesting, and I recommend publication.